# Measuring Rural Food Environments for Local Action in Australia: A Systematic Critical Synthesis Review

**DOI:** 10.3390/ijerph16132416

**Published:** 2019-07-07

**Authors:** Penelope Love, Jillian Whelan, Colin Bell, Jane McCracken

**Affiliations:** 1Institute for Physical Activity and Nutrition, School of Exercise and Nutrition Sciences, Deakin University, Waurn Ponds 3216, Victoria, Australia; 2Global Centre for Obesity Prevention, School of Health and Social Development, Deakin University, Geelong 3220, Victoria, Australia; 3Northern Mallee Community Partnership, Mildura 3500, Victoria, Australia

**Keywords:** rural, regional, food environments, measurement tools

## Abstract

Poor diet is a significant contributor to obesity and chronic disease. With all being more prevalent in rural than urban Australia, modifying the food environment is a potential intervention point to improve the health of rural populations. This review examined the applicability of measurement tools used in rural food environment research for rural Australia. Six electronic databases were searched for peer-reviewed literature, published in English between 2006 and 2018, including at least one objective measure of the Community or Consumer Food Environment in a rural or mixed rural/urban context. One-hundred and seventy-seven papers were returned after removal of duplicates, with a final review of 25. Most studies were cross-sectional, with one intervention study of quasi-experimental design. Nine studies employed a conceptual model; there was considerable variability in tools used; and few described psychometric testing. The most common attribute measured was price, followed by available healthy options. The findings of this review do not offer a suite of ‘gold standard’ measurement tools known to be reliable, valid and sensitive to change to assess the community or consumer food environments in rural Australian towns. However, recommendations are proposed to progress this important area of research within a rural context.

## 1. Introduction

Poor diet is a leading risk factor for preventable disease both in Australia and internationally [1,2]. In high-income countries, obesity affects all age groups, disproportionately affecting those from more disadvantaged backgrounds, while in low- and middle-income countries the obesity burden begins with the well-off and shifts to the rural poor as a nation’s gross domestic product increases [3]. Fewer than 7% of Australian adults currently consume a diet consistent with the Australian Dietary Guidelines [4], with dietary risks (11%) and overweight and obesity (9%) the leading contributors to the burden of disease in Australia [5]. This disease burden is not shared evenly across Australia with those in geographically remote areas at greater risk [1]. While overweight and obesity affects 61% of adults in major cities, it affects 69% of adults in outer regional and remote areas. Similarly, the prevalence of diabetes in major cities is lower (4.7%) compared to outer regional/remote areas (6.7%), and cardiovascular disease is lower (4.7%) in major cities compared to outer regional/remote areas (5.8%). Nationally, a 4% reduction in total disease burden could be achieved if all Australians experienced the same level of health as those living in a major city [1]. 

Behavioural risk factors alone cannot adequately explain the global population level rise in obesity in the last three to four decades [3] and there is growing recognition that environmental factors play a role [6,7]. The food environment can be defined as “...the collective physical, economic, policy and socio-cultural surroundings, opportunities and conditions that influence people’s food and beverage choices and nutritional status” [7]. Prioritising interventions that target the food environment, such as improving the availability and affordability of healthy foods, rather than programs aimed at individuals, has the potential to improve population-level diet quality in an equitable manner [8].

Food environment research is a growing field of academic endeavour, with some evidence that healthier food environments are associated with healthier diet quality [9]. However, there remains much uncertainty about the pathways through which food environments influence dietary intake, obesity and chronic disease [10]. Numerous measures of the food environment exist, but there is little consistency in their application and few tools have undergone rigorous psychometric testing to ensure reliability and validity [11]. The development and use of standard methodology is important for the collection and comparison of data locally, nationally and globally to inform the implementation of appropriate interventions. A standardised approach is also supported by the International Network for Food and Obesity/non-communicable Diseases Research, Monitoring and Action Support (INFORMAS) [12].

To progress our understanding of the role that food environments play in dietary patterns and obesity, numerous researchers propose the use of conceptual models [13,14,15] to build on previous findings and to further an understanding of model constructs [16]. The majority of recent systematic reviews examining the relationship between the environment and obesity use the widely accepted socio-ecological conceptual model proposed by Glanz et al. [17], suggesting that it is the most useful way of organising food environment constructs for research and practice [18]. The model is applied at community and consumer levels, and captures how policy, environmental and individual level variables interact to influence eating behaviours. Four broad subtypes of environmental variables are described: Community, Organizational, Consumer and Information [17] (Figure 1).

The community food environment encompasses the number, type, location and accessibility of food outlets; and the consumer food environment is defined as “what consumers encounter within and around a retail food outlet” and includes aspects of price, promotion, placement, nutrition information and available healthy options [17]. In subsequent research, Glanz and colleagues [19] included quality as an aspect of the consumer food environment, supported by numerous other authors [10,18,20,21], especially in a rural context [22,23]. The organizational food environment includes school canteens, worksite food outlets, healthcare facilities and homes; and the information environment includes advertising and media operating within retail environments, at a local neighbourhood or a national level [17]. 

Glanz and colleagues [17] argue that the community and consumer food environments have a broad impact at a population level and are therefore a high priority for researchers. Determining the most appropriate methodology to measure these two environments in rural Australian towns is therefore the focus of this review. It is not within the scope of the review to explore the measurement of the information or organizational food environments. 

This review seeks to answer the following research question: **Are the measurement tools employed in rural food environment research appropriate to inform local interventions in rural (outer regional and remote) Australian towns?** Within the Australian context, outer regional and remote towns are categorised as having an ARIA (Accessibility Remoteness Index Australia) score of >2.4–5.92 and >5.92–10.53 respectively, indicating a measure of road distance between populated localities and service centres [24].

Numerous literature reviews focus on the methods and measures available to assess aspects of community and consumer food environments, such as Glanz et al. [10], Kelly et al. [21], Caspi et al. [25] and Lytle et al. [26]. To the best of our knowledge, this is the first review worldwide to focus specifically on the appropriateness of measurement tools to assess rural community and consumer food environments.

## 2. Methods 

This review follows the Preferred Reported Items for Systematic Reviews and Meta-Analyses PRISMA guidelines [27]. The protocol for this review is registered on PROSPERO (registration: CRD42018116003).

### 2.1. Search Strategy

Key words identified in systematic reviews undertaken by Caspi et al. [25] and Glanz et al. [10] were used to identify search terms. The term “rural*” was added to the search string being specific to the topic of this review. The final search string used was “food access” OR “food availability” OR “food quality” OR “food affordability” OR “food cost” OR “food price*” OR “food promotion*” OR “food placement” OR “food environment” OR “nutrition environment” OR “Community Food Environment” OR “Consumer Food Environment” OR “Community Nutrition Environment” OR “Consumer Nutrition Environment” AND measure* OR assess* OR research AND rural* OR region* OR remote AND Australia OR Canada OR USA OR “United States of America” OR NZ OR “New Zealand” OR UK OR “United Kingdom” OR Scotland OR Ireland OR Wales OR England. Six electronic databases were searched using the EBSCOhost platform: Academic Search Complete, CINAHL Complete, Global Health, Informit, Medline Complete and PsychINFO. 

### 2.2. Inclusion Criteria

Peer-reviewed literature was included if: the paper presented original research findings on the development/use of at least one objective measurement tool or method to assess any aspect of the community or consumer food environment; at least a component of the research was conducted in a rural context; the research was undertaken in a high-income country comparable to Australia; the research was reported in English; the research was published between 2006 and 2018. The field of research has progressed significantly in the last decade [10], therefore research conducted prior to 2006 was deemed unlikely to be relevant to the research question. Studies in countries comparable to Australia were included given the scarcity of Australian studies and their key focus on food pricing. 

### 2.3. Exclusion Criteria

Peer-reviewed literature was excluded if it: was published prior to 2006; did not include any data collection from rural areas; was published as part of conference materials, a student thesis or dissertation; did not include sufficient detail about the objective measurement tool/s or method; was focused on a specific ethnic group; was spurious to the research question. The exclusion criterion regarding specific ethnic groups was added when the initial search returned a number of studies in very remote, isolated locations reviewing Aboriginal Australian and American Indian Reservation community food environments; rather than outer regional, remote areas. 

### 2.4. Search Strategy Results

The initial search returned 252 studies; 177 studies remained after duplicates were removed. Abstract screening excluded a further 146 studies, leaving 31 studies for which the full text version was accessed and assessed for eligibility. A further 10 studies were excluded because the research: did not include rural environments; was undertaken in an environment specific to a particular ethnic group; did not describe the objective measurement or primary data collection. The reference lists of the final selection of studies were hand searched for additional papers meeting the inclusion criteria, with the addition of four studies. Screening of titles, abstracts and full text articles was done by JM, JW and PL; with any uncertainties resolved through consensus by PL and CB. The final review contained a total of 25 studies (Figure 2).

### 2.5. Data Extraction and Analysis

A data extraction table was created based on the work of Glanz et al. [10] and Gustafson et al. [28], and used to systematically extract information from each study relevant to the research question: geographical setting (rural or mixed urban/rural); food environment component studied; aspect/s of food environment measured; tools/methods used; psychometric characteristics; and context (number and mix of retail food outlets measured). Data was extracted by JM, JW and PL, then verified by all authors (Appendix A). 

The STROBE Checklist (for cross-sectional and cohort studies) [29] and the TREND Statement (for the quasi-experimental study) [30] were used to critically appraise each study. Two authors independently appraised each study, with consensus undertaken by a third author (Appendix A). 

## 3. Results

### 3.1. Description and Quality Appraisal of Studies

In assessing the methodological quality of included studies, seven studies rated as moderate and 18 as strong (Appendix A). Studies performed poorly in relation to sources, sample size and unit of analysis, selection bias, collection of baseline data, management of missing data, and reporting on limitations, especially unintended consequences.

Of the 25 studies, 11 were conducted in Australia, 9 in the USA and the remainder in Scotland (2), Canada (2) and New Zealand (1). Thirteen studies were conducted in a completely rural context with the remaining 12 conducted in mixed urban/rural environments. Fifteen studies provided a definition [or reference to a definition] of rurality. Nine studies referred to a conceptual model, all referencing the conceptual model proposed by Glanz et al. [17]. One study, by Martinez-Donate et al. [31], was an intervention study [of quasi-experimental design] with the remainder being observational [20 cross-sectional; 4 longitudinal cohort]. Although Pereira et al. [32] and Pitts et al. [33] have been included as observational cross-sectional studies, their results were intended to inform quasi-experimental interventions. Glanz et al. [10] and Lytle et al. [26] note the importance of undertaking rigorous psychometric assessments of the tools used to measure the community and consumer food environments in this relatively new area of research. The psychometric characteristics of measurement tools were discussed in 11 studies, with inter-rater reliability, test–re-rest reliability, and validity most commonly described (Table 1).

### 3.2. Food Environments Investigated by Studies

Employing a mix of assessment techniques across both the community and consumer food environments is considered vital to accurately depict the complexity of food environments and to identify multiple points for potential intervention [10,25]. Seven of the reviewed studies described both food environments [40,46,47,48,49,50,52]; two studies described the community food environment only [39,53]; and the remaining 16 studies described the consumer food environment only (Table 2).

### 3.3. Community Food Environment

Nine studies measured aspects of the community food environment; the most common being type and location of food outlet (8 studies) [39,40,47,48,49,50,52,53], followed by accessibility of food outlet (4 studies) [46,48,49,52]. Three studies [40,46,52] employed a conceptual model. Four studies [40,46,48,52] reported on the reliability or validity of the tools/methodologies employed. There was wide variability in the methods used in terms of how mapping was undertaken and scales adopted.

#### 3.3.1. Type and Location of Food Outlet

Sharkey et al. [47] used a two-stage process to determine the type and location of fast food outlets, and access to traditional fast food and healthier fast-food in six rural counties in Texas, USA. Stage one involved trained observers systematically driving all roads in the county using a Global Positioning System (GPS) to plot all traditional food stores such as supermarkets, grocery stores and convenience stores, and all non-traditional food stores such as dollar stores, mass merchandisers and pharmacies. Stage two involved determining which of the 261 outlets sold healthier or traditional ‘unhealthy’ fast food options using an in-store checklist. Spatial access to fast food was calculated by determining the proximity of outlets selling traditional fast food and healthier fast food options by road networks to populated areas or ‘Census Block Groups’ (CBGs). Coverage of outlets [number of fast food purchasing opportunities in a defined area] was calculated in general and for healthier fast food options within one, three and five miles of the CBGs, but no explanation was provided as to why these distances were chosen. The authors found that mapping only traditional fast food outlets, such as take-aways, in rural areas significantly under-estimated neighbourhood exposure to unhealthy fast food [47]. Creel et al. [49] found that 59% of the opportunities to procure fast food in a rural setting were outside of traditional fast food outlets.

Sharkey et al. [47] also determined that retail food outlets should be mapped through ‘ground-truthing’; physically viewing and recording outlet locations and not relying solely on a commercial business listing or other secondary source. Using secondary sources to obtain information on the location of retail food outlets is considered acceptable practice, however this technique should be used with caution [55,56] as described by Innes-Hughes et al. [40] who found only a 25–39% agreement between the directory of commercial businesses and direct observation of retail food outlets across the three towns they studied. 

Tseng et al. [39] studied the relationship between the environment and the body mass index (BMI) of urban and rural women, mapping fast food chain stores within two kilometres of each participant’s residence. They describe a 2 km residential buffer zone as ‘arbitrary’ and suggest the collection of a number of indicators at a neighbourhood level. When exploring the food environments of primary school children (aged 3–12 years) in rural Canada, DuBreck et al. [52] categorised “junk food opportunities” as including convenience stores, grocery stores and fast food outlets (casual and full service). Based on the school zone, 800 m and 1600 m walking network buffers were measured around each school; rather than circular buffers, which might include barriers to walking such as rivers or railway tracks. These two distances were chosen as 800 m is recognised as being walkable by children in 10–15 min, and 1600 m is the school-board mandated distance before a student is eligible for the school bus service.

Sadler et al. [53] used Global Information System (GIS) mapping to measure the community food environment in rural Canada. Rather than mapping from a CBG, they mapped via road from individual households directly to a variety of store types, increasing the accuracy of the measurement. The use of driving distance to determine accessibility to outlets rather than walking distance [which is a feature of many urban studies] is justified in that this is how most rural residents obtain food [53]. They also included food outlets beyond municipal boundaries in consideration of how rural communities engage in ‘out-shopping’ [57] beyond their immediate boundaries. Sadler et al. [53] therefore argue that reliance on store type as a proxy for the food available within the outlet is likely to result in an underestimation of access to fast food. 

Hosler et al. [48] and Smith et al. [50] both undertook spatial accessibility using GIS mapping, exploring and comparing urban and rural food environments in the USA and Scotland respectively. Hosler et al. [48] GIS mapped 263 retail outlets identified as stocking fruit and vegetables, calculated store density per 10,000 residents to estimate fruit and vegetable availability in different municipalities, and then calculated a weight-adjusted density taking into consideration store size and business hours. Smith et al. [50] determined the availability of fruit and vegetables through visiting stores and using a Consumer Nutrition Environment checklist. Similar to Sharkey et al. [47], Smith et al. [50] used CBGs as the starting point to measure access to the closest fruit and vegetable source, however they used road-based travel time rather than distance to determine access, making comparison with other studies difficult. The authors found poorer spatial accessibility to fruit and vegetables in deprived neighbourhoods in island, rural and small town settings [50].

#### 3.3.2. Accessibility of Food Outlet

Accessibility, defined as encompassing store opening hours and drive through access [17], was assessed in three studies [46,48,49]. Only one study [49] made use of this information, undertaking a weighted calculation of fruit and vegetable access using store size and opening hours. 

### 3.4. Consumer Food Environment

The two most commonly researched aspects of the consumer food environment were price (16 studies) [22,31,32,33,34,35,36,37,38,41,42,43,44,45,52,54] and available healthy options (15 studies) [31,32,33,36,40,43,44,45,46,47,48,49,50,52,54], supporting the findings of Glanz et al. [10]. The next most commonly researched aspect was quality (10 studies) [22,31,32,33,36,43,44,45,46,51]; with promotion, placement and nutritional information researched by three studies [31,32,52] (Table 3). The psychometric properties of the consumer food environment tools used in the reviewed studies are presented in Table 3. 

#### 3.4.1. Price of Healthy Options

Within this review, the most common method to collect food pricing information was through a ‘healthy food basket’ using a pre-defined list of foods in quantities representative of the total diet of referent families over a defined period of time. Six different healthy food basket methodologies were used: the Victorian Healthy Food Basket (VHFB) [37,41,42]; the Queensland Healthy Food Access Basket (QLD HFAB) [38], also used as the basis for the Western Australian tool [22]; a healthy food basket for New Zealand [54]; a pilot tool tested in rural and urban Tasmania [43]; and the Healthy Diets Australian Standardised Affordability and Price (ASAP) tool [35]. All food pricing studies using a healthy food basket methodology were from Australia and New Zealand.

The Victorian Healthy Food Basket (VHFB) [58] was developed in 2007 and meets 80% of nutrient requirements and 95% of energy requirements for a two week period for four different referent families; comprises 44 items from the five core food groups and a chocolate bar and soft drink for comparison purposes; and can be compared across stores, towns or regions. In their cross-sectional study using the VHFB in a convenience sample of 34 supermarkets in rural Victorian towns, Palermo et al. [42] did not find an association between the price of a healthy food basket and remoteness, socio-economic index, population size, density or distance from Melbourne. In contrast, Ward et al. [41], using the VHFB in their cross-sectional study of 14 supermarkets in 10 rural South Australian towns and 61 metropolitan supermarkets in Adelaide, found the cost of a healthy food basket higher in rural areas, although not statistically significant given the small sample of rural supermarkets included in the study. Interestingly, in their 3-year longitudinal study in Victoria (2012–2014) using the VHFB, Palermo et al. [37] found that the distance of a retail food outlet from Melbourne was predictive of increased costs. 

Two methodological limitations of the VHFB are that the basket does not include generic brand items, which are becoming increasingly prominent on Australian supermarket shelves and the exclusion of stores if they contain fewer than 90% of the 44 items [62,63]. There is no mention of reliability or validity testing of the VHFB in any of the studies using this methodology, nor in the development of the tool [58].

In contrast to the VHFB, the Queensland Healthy Food Access Basket (QLD HFAB) methodology [59] collects the price of the lowest cost brand, regardless of whether it is a generic brand, and does not exclude stores if they stock less than 90% of the items in the basket. The QLD HFAB includes commonly available and popular items from the core food groups in the 2003 Australian Dietary Guidelines, and is designed to meet 95% of the energy requirements and 70% of the nutritional requirements for a reference family of six for a fortnight. A meat pie, a cola beverage and two tobacco products are also included in the basket as ‘unhealthy’ items for comparison. The QLD HFAB methodology was adapted by Chapman et al. [38] for their 3-year longitudinal study in NSW, who found that the overall cost of a healthy food basket was more expensive in remote compared to highly accessible areas of NSW, especially fruit and vegetable costs. Between 68 and 95 surveyors collected the data each year with no information regarding surveyor training prior to data collection; limited inter-rater reliability testing was undertaken; and no reference is made to any test–re-test reliability or validity testing [38].

The QLD HFAB methodology was adapted by Pollard et al. [22] for the Food Access and Costs Survey (FACS) conducted by the Western Australia (WA) Department of Health. Their study of a stratified random sample of 144 retail outlets found that the cost of a healthy food basket significantly increased as distance from major cities increased. They also found that the price of a healthy food basket was higher in remote and very remote areas of WA [22]. Although the study used a cross-sectional study design, a number of notable strengths in the methodology are apparent, namely: all surveyors undertook extensive training prior to administering the tool; the survey was conducted in the same time period across all stores to minimise confounders; and the sample size (n = 144) was representative of the total number of supermarkets in WA (n = 447). The study made use of 97 surveyors to collect data but inter-rater reliability testing is not described [22].

Healthy food basket methodologies used by Wang et al. [54] and Herzfeld et al. [43] show poor design and implementation, with the healthy food basket items not based on consumption data or dietary recommendations [54] and low scores for inter-rater reliability [43].

Given the limitations that the “healthy food basket” methodology poses for small rural towns, Love et al. [35] applied the recently developed Healthy Diets Australian Standardised Affordability and Price (ASAP) tool. This tool assesses the price, price differential and affordability of the recommended Australian diet (as defined by the Australian Dietary Guidelines) with the current Australian diet (as measured by the Australian Health Survey) using the reference household of two parents and two children (boy aged 14 years; girl aged 8 years). Price is collected for 43 food items, adjusted for edible proportion, representative of the recommended Australian diet (fruit; vegetables and legumes; grain/cereal foods; meats, poultry, fish and alternatives; milk, yoghurt, cheese and alternatives; unsaturated oils and spreads), and an additional 33 ‘discretionary’ food items, high in saturated fats, sugars, salt and/or alcohol. Using the Healthy Diets ASAP tool, Love et al. [35] confirmed that while purchasing a recommended (healthy) diet may be less expensive than the current diet, it would account for almost a third of the budget for median and low-income households in rural Victoria, Australia. The authors describe the Healthy Diets ASAP tool as a practical and time efficient survey, and make several recommendations for its use in a rural context [35]. 

The Nutrition Environment Measures Survey for Stores (NEMS-S) and Restaurants (NEMS-R), or variations of these tools, were used to assess price in five reviewed studies [31,32,33,36,46]. Rather than collecting the actual price of a food basket, the NEMS-S and NEMS-R tools compare the price between healthy and unhealthy options of particular foods to determine potential price differences in stores and restaurants. Two of these studies [32,36] specifically reported on their findings in relation to pricing. Whelan et al. [36] reported that healthier options were usually more expensive and less available across their sample of six general stores and five supermarkets, especially low fat dairy products and wholegrain breads/cereals. Pereira et al. [32] reported mixed findings in their small sample of three grocery stores and five convenience stores when comparing the price of healthier and less healthy alternatives, with some healthy options being cheaper (such as low fat milk and high-fibre cereal) and some being more expensive (such as lean mince and whole-wheat bread). They noted the difficulty of comparing standard products and the healthier alternative in convenience stores due to many convenience stores not stocking the healthier product option. 

#### 3.4.2. Availability of Healthy Options

The NEMS-S and NEMS-R, and modifications of these instruments, were the two most commonly used tools to measure available healthy options. Martinez-Donate et al. [31], Pereira et al. [32] and Whelan at al. [36] were the only studies to use the NEMS-S and NEMS-R tools according to their respective protocols [19,64]. The NEMS-S contains a list of 10 food products, each with alternative healthier options, as well as fresh fruit (10 types) and fresh vegetables (10 types). The 10 food products were based on US Federal Government guidelines at the time the tool was developed [19]. 

Andreyeva et al. [65] adapted the NEMS-S to reflect the foods that they felt were important in a healthy low cost diet. The revised tool, NEMS-S-Rev, includes 12 food categories and healthier options as well as 10 types of fresh fruit, 10 types of fresh vegetables, frozen vegetables (peas, broccoli and green beans), and canned vegetables (corn and green beans) [65]. Pitts et al. [33] used the NEMS-S-Rev in their study of 33 corner stores and nine supermarkets, justifying its use in terms of canned items being more frequently purchased by low-income people in rural areas who may live some distance from stores that carry fresh items. Results indicated that rural corner stores provided more healthy options than urban corner stores. The reliability and validity of the NEMS-S-Rev was not discussed in either study [33,65].

In Australia, the NEMS-S tool was modified for use in rural towns by Innes-Hughes et al. [40] and Whelan et al. [36]. Innes-Hughes et al. [40] developed a rapid assessment tool including healthy indicator foods [reflecting the Australian Guide to Healthy Eating core food groups], and unhealthy indicator foods (reflecting popular and commonly consumed energy-dense nutrient poor foods/beverages). The authors reported excellent inter-rater and test–re-test reliability [40]. Whelan et al. [36] converted imperial measures to metric, substituted hotdogs with chicken (with and without skin), used Australian reference brands, and included seasonal fruits and vegetables. They pilot-tested the modified survey for face validity prior to use and reported a high level of test–re-test reliability during data collection [36].

The Farmers Market Audit Tool (F-MAT) developed by Byker et al. [46] is also based on the NEMS-S; developed in response to a gap in research methodology concerning the availability of food types at farmers markets. Face validity, inter-rater reliability and discriminant validity of the tool were all examined during pilot testing of the tool in rural and urban environments in three US states. The F-MAT was found to be a reliable and valid way to assess food quality and availability at farmers markets [46]. 

The NEMS-R tool was developed as an observational measure for use in restaurants and fast food outlets to assess available healthy options, facilitators and barriers to healthy eating, pricing, and signage/promotion [64]. The NEMS-R was used in one study in Australia [36], and two studies in the USA [31,32]. Whelan et al. [36] surveyed all (n = 27) food service outlets across a rural local government area in Victoria, Australia; Martinez-Donate et al. [31] surveyed seven restaurants and one fast casual restaurant in two rural towns in Mid-Western USA; and Pereira et al. [32] surveyed 34 restaurants in the rural town of New Ulm, Minnesota. Although not specifically stated, it would appear that Innes-Hughes et al. [40] used the NEMS-R as a basis for the development of their rapid assessment tool for take-away food outlets in NSW, Australia. The NEMS-R tool is valid in an American context, providing acceptable inter-rater reliability and very good test–re-test reliability [64]. A concern raised by Whelan et al. [36] is the higher scoring of promotion and signage, rather than availability, within the NEMS-R tool.

#### 3.4.3. Quality of Healthy Options

Ten studies [22,31,32,33,36,43,44,45,46,51] assessed the quality of fresh fruit and vegetables in corner stores, grocery stores and supermarkets using five different methodologies. Pollard et al. [22] provide the most detail about their methodology, and are the only authors who give an operational definition of quality: “…a perception of freshness (based on) appearance, colour, aroma/odour, texture, size, shape, flavour and freedom from defects”. They undertook an extensive literature review and developed quality criteria for 13 commonly consumed fruit and vegetables using wording adapted from the Australian Industry Horticultural Quality Grading System [22]. Their study of 144 food retail stores across WA found the quality of fresh fruit and vegetables was lower in remote supermarkets and stores, identifying this as a possible intervention point to improve diet in rural communities. Their study undertook minimal inter-rater reliability testing, with no mention of test–re-test reliability or validity testing [22].

Focusing on the quality of fruit and vegetables in 288 food stores in 10 communities across Scotland, Cummins et al. [51] adapted the quality assessment aspect of the Healthy Eating Indicator Shopping Basket (HEISB) [60]. The adapted tool consists of a visual assessment based on a 3-point Likert scale (1 = poor, 2 = medium and 3 = good), accompanied by a detailed description of how to interpret the scale for four fruit and eight vegetable types [51]. The authors found the overall quality of produce was high in both urban and rural stores, but tended to be lower in stores where fresh food was a secondary item (such as corner stores). The authors note that the 3-point Likert scale “…may not have been sufficiently sensitive to capture the full range of variation in item quality” and highlight the subjective nature of the tool [51]. Psychometric testing was not described in the development of any aspect of the HEISB tool either [60].

The NEMS-S tool uses a binary scale of ‘acceptable’ and ‘unacceptable’ for 10 fruit and 10 vegetable types “…based on the majority of [the] given type of fruit or vegetable being clearly bruised, old looking, over-ripe or spotted” [19]. Although used by Martinez-Donate et al. [31], they do not specifically report any findings on the quality of produce. Pereira et al. [32] found that quality was lower in convenience stores. Whelan et al. [36] reported a lack of variability in quality scores, recommending that the protocol be made more sensitive for scores >75% but less than 100%.

The NEMS-S quality rating system for produce has been integrated in the F-MAT tool [46], however the reliability and validity of this aspect of the tool is not specifically examined; and while the NEMS-S-Rev [65] is based on the NEMS –S, the methodology adopted to assess quality of fresh produce is quite different. Using the NEMS-S-Rev in 33 corner stores and nine supermarkets in North Carolina, Pitts et al. [33] scored the quality of 12 fruit and 12 vegetable types using a 3-point Likert scale of acceptability in terms of bruising, discoloration and rotting (1 point = 25–49% acceptable; 2 points = 50–74% acceptable; 3 points = 75% or more acceptable). The authors concluded that the quality of produce was lower in rural compared to urban areas overall, but for rural corner stores the quality of produce was similar to urban corner stores [33].

Herzfeld and McManus [43] was the only reviewed study to suggest the collection of contextual information including date of fruit and vegetable delivery to the retail outlet; storage conditions; source of produce; operating structure; and location to enhance observational fruit and vegetable quality data. No recommendations were made on how this information should be used however.

#### 3.4.4. Promotion, Placement and Nutritional Information

These aspects were assessed in three studies, two of which used the NEMS-R tool [31,32]. Martinez-Donate et al. [31] found that NEMS-R scores improved overall in the seven restaurants involved in their intervention, mainly due to changes in signage, identification of and promotion of healthier foods. Using the NEMS-R tool in 34 fast food and sit-down restaurants, Pereira et al. [32] found mixed results, with fast food restaurants more likely to have nutrition information available for customers at the point of purchase, less likely to offer an ‘all you can eat option’, and more likely to have signage encouraging larger portions, than sit-down restaurants. Although Creel et al. [49] state that data regarding nutrition information was collected in their study of 205 rural traditional and non-traditional fast food outlets, no results or methodological information are provided. One study [52] used an expanded version of the NEMS-R tool, the Children’s Menu Assessment (CMA) tool, developed to assess children’s restaurant menus in more detail.

### 3.5. Measuring Food Environments in Intervention Studies

Martinez-Donate et al. [31] was the only intervention study, focusing on all six aspects of the consumer food environment, and designed as a pilot to inform future community-based interventions in rural areas. The NEMS-S and the NEMS- R tools were used to obtain baseline and post-intervention (10 months after implementation) measures in a quasi-experimental study in two rural US communities, one intervention and one control (30 miles apart). Seven restaurants and two supermarkets were involved in the intervention community, matched with comparator supermarkets and restaurants in the control community. Each participating outlet chose and agreed to implement a minimum of three strategies from a pre-defined list. Average NEMS-R scores improved in intervention restaurants versus the comparison community restaurants; and average NEMS-R scores increased in the intervention restaurants. Detailed statistical analysis was not undertaken due to the small sample size [31]. Based on these results, the authors state the NEMS-R is a suitable tool to measure intervention effects in restaurants but question whether the NEMS-S is a suitably sensitive tool to detect an intervention effect in a supermarket environment. The authors also suggest that audit tools measuring interventions to address the consumer food environment in supermarkets should include the measuring of marketing and promotional practices [31]. While the study had notable strengths of the study design, including data collectors being blinded to the study purpose and the inclusion of a control community, the study intervention and control communities were not equally matched in terms of population size [31]. 

## 4. Discussion

This review highlights the limited number of tools that exist for evaluating rural food environment interventions, despite evidence of higher rates of diet-related disease burden in these communities in Australia and internationally. Of the measurement tools assessed in this review, none were appropriate in their current form to describe or inform local interventions to improve the community or consumer food environments in rural (outer regional and remote) Australian towns. 

This review also exposes a general lack of rigorous psychometrically tested tools available to measure the community and consumer food environments. This issue is not confined to rural food environment research, but is reflective of the broader field, with Glanz et al. [10] reporting 50.4% of their reviewed studies examining the reliability of tools, 30.4% testing for validity and only 13.6% investigating sensitivity to change. Other researchers have found similar results [18,25,28], however, no clear benchmarks are available, against which the reliability, validity and sensitivity of food environment tools can be assessed [14,26,66].

In their recent systematic review of available measures of the consumer food environment, Glanz et al. [10] note that of the 130 included studies, 85.6% were observational cross-sectional studies. This review produced similar results, with 24 of the 25 reviewed studies being observational and 20 cross-sectional. This preponderance of observational studies poses questions as to the purpose of collecting food environment information, and how this information is used to inform local, community level interventions. Of particular relevance to this review was the ability of the tools to detect change. With only one intervention study included, it is difficult to recommend food environment measures or tools for this purpose, as their ability to detect change is unknown. Martinez-Donate et al. [31] question whether the NEMS-S is sensitive enough to detect change in supermarket-based intervention studies, arguing that the tool does not measure promotion and placement which are amenable to intervention and are known to impact on consumer food choices [67,68]. Robust evidence about the ability of existing tools to measure change in food environments following interventions is acknowledged as a current gap in the research due to the lack of intervention studies [10].

In addition to the use of valid and reliable measurement tools which are sensitive to change, it is recommended that both community and consumer food environment measures be undertaken to obtain a clear and detailed picture of the food environment in a particular locality [11]. Caspi et al. [25] describe this as the single most important strategy for future research. It is also suggested that researchers “…explicitly employ hypothesized causal models to link environmental features with diet-related diseases” [16,69] to build on previous findings and enhance understanding of the associations between nutrition environments, dietary patterns, obesity and chronic disease. Only seven reviewed studies undertook measurement across both environments. Similarly, only nine reviewed studies employed a conceptual model, with Glanz’s Conceptual Model of Nutrition Environments [17] most commonly used. 

### 4.1. Community Food Environment (Type, Location and Accessibility of Food Outlets)

Tools/methodologies used to measure the community food environment were highly variable; with nine reviewed studies adopting different approaches; none being intervention studies; and reliability or validity of the methods employed was seldom assessed. This is reflective of the broader field of study, which is hampered by inconsistent assessment methodologies [12]. Evidence of an association between the community food environment and BMI or dietary intake is inconclusive at this stage, largely due to the wide range of research methodologies employed and the inability to make comparisons between studies [70].

This review did not reveal a ‘gold standard’ measurement tool known to be reliable, valid and sensitive to change to assess the community food environment in rural Australian towns. However, it did identify features that would be important for such a tool. The use of network/buffer zones and considerations of access in rural studies should be at the scale of the car; not walking distance as is used in most urban studies [53]. Both traditional and non-traditional fast food outlets should be included when researching rural environments as mapping only traditional fast food outlets can result in an under-estimation of access to unhealthy food [47,49]. This highlights the importance of combining community and consumer food environment measures and perhaps ranking [71] the complex local food environment, rather than dichotomising it into ‘healthy’ and ‘unhealthy’ outlets based solely on outlet type [21,25]. Mapping outlets beyond municipal boundaries may reduce underestimation of food access and more closely reflect actual food purchasing behaviours of rural communities [53,57]. Using secondary sources to collect information on outlet location is acceptable practice but may be enhanced by ‘ground-truthing’ [40,47]. The appropriateness of collecting information seldom used, such as accessibility data, should also be addressed [14]. 

### 4.2. Consumer Food Environment (Price, Availability, Quality, Placement, Promotion and Information)

A range of complex and inter-related factors including politics, economics and culture interact at local, national and international levels to influence the price of food [72]. While monitoring the price of food is certainly important, such research findings are more likely to inform State and National efforts rather than local level interventions considering the complex set of influencing factors. Price monitoring and intervention efforts need to adopt consistent methodology which allows comparison between different geographical areas and population cohorts to have impact on policy [62]. This review demonstrates the considerable heterogeneity in healthy food basket methodology employed in rural Australian and New Zealand research, with 6 different methodologies included in this review, and at least 11 different methodologies known to exist in Australia [72]. Evidence suggests that price differs according to geographical remoteness in Australia [35,37,62,72] therefore monitoring food price in a uniform manner across Australia, and stratifying according to remoteness and possibly distance from the State capital city, would produce useful information. The recently developed Healthy Diets ASAP tool will facilitate this [61,72].

Measurement of the promotion and placement aspects of the consumer food environment may be useful to inform local level interventions in stores and restaurants through healthy signage and the identification and promotion of healthier foods. Supermarkets represent a key setting for food purchasing in Australia [73] and play a crucial role in shaping population level diets [74]. They are therefore critical settings for interventions in rural Australian towns [75]. Cameron et al. [68] have found shelf labelling to be a promising intervention to increase the purchase of healthier foods in Australian supermarkets, and there is some international evidence to suggest that the prominence of placement of items and amount of shelf space impacts consumer purchasing patterns [12,76]. A measurement tool sensitive to the promotion and placement aspects of the consumer food environment is likely to be important for rural towns, such as the *GroPromo* audit tool used to assess placement of seven product categories in promotional locations in large chain grocery stores in the USA [77]. The tool is reliable and valid in an American context and a slightly modified version is recommended for the Australian context [6]. 

With the implementation of menu board disclosure (mandatory kilojoule labelling laws) across parts of Australia, measuring the nutrition information aspect of the consumer food environment will become more important. This is especially so for smaller rural towns where large fast food chain outlets may not exist, but numerous privately owned fast food outlets, not covered by the legislation, do [47,49,78]. This requires monitoring using standard measures, ideally using an adapted version of the NEMS-R for the Australian context. 

There is a gap in the evidence base concerning the impact of quality on food choice and how to measure this aspect of the consumer food environment [25]. In their original conceptual model, Glanz et al. [17] did not include quality but referred to its inclusion in later publications [19,20]; and recent adaptations of the conceptual model now include quality as a key attribute [18]. This review did not identify a valid or reliable tool to measure quality, however Pollard et al. [22] provide the most comprehensive detail on how this attribute could be assessed in Australia, but would require psychometric testing for the rural context. 

### 4.3. Limitations 

Definitions of rurality varied widely between and within countries, and between authors. This limitation may have been minimised by only including Australian studies, however this would have narrowed the scope of the review to the aspect of food pricing. The inclusion of International studies does raise the question of transferability of results to the Australian context. 

This review specifically focused on the use of measurement tools to inform interventions at a local, community level, but with only one intervention study identified it is not possible to make conclusive recommendations. This is reflective of the broader field of food environment research, where intervention studies are just emerging. Recommendations are made as a means of furthering the field of rural food environment research, although these should be interpreted with caution.

This review would have benefited from the application of a critical appraisal tool designed to systematically examine the psychometric properties of these tools. After an extensive search and correspondence with leading academics in the field, such a tool could not be found. Therefore, a data extraction table was developed based on the recent work of Glanz et al. [10] and their review of the psychometric properties of food environment measurement tools (Appendix A). The STROBE Checklist [29] and TREND Statement [30] were applied to assess the quality of the reviewed studies, however these tools are intended for individual rather than environmental level studies making some questions difficult to apply. 

Only nine reviewed studies applied Glanz’s Conceptual Model of Nutrition Environments [17], requiring JM and PL to categorise the remaining studies according to their interpretations of these studies. There is potential for bias in this method. 

### 4.4. Proposed Recommendations for Measurement of Food Environments within a Rural Context:

Community Food Environments:GIS mapping should be undertaken at the scale of the car, from individual households directly to a variety of store types, increasing accuracy of measurementTraditional and non-traditional fast food outlets should be mapped to avoid under-estimation of access to unhealthy foodOutlets beyond municipal boundaries should be mapped to reflect how people interact with their food environmentMapping of outlet location should be based on ground-truthing; not solely on secondary sourcesAccessibility data should only be collected when there is a clear purpose to do soThe INFORMAS protocol to investigate food retail outlets [12] may provide an opportunity for the standardised collection of community food environment data across Australia

Consumer Food Environments:Standardised methodology for the collection and comparison of pricing information is recommended using the Healthy Diets ASAP tool [61,72]Measurement of the promotion and placement of food may be most useful to inform local level interventions, and an adapted version of the *GroPromo* audit tool [77] may be suitable for this [6]An adapted version of the NEMS-R is recommended for prepared food outlets to assess the impact of nutrition information, placement, available healthy options and priceFurther psychometric testing is vital to develop a more robust food quality assessment tool

## 5. Conclusions

A better understanding of the food environment, and the role it plays in influencing dietary patterns, is needed to address the inequitable burden of overweight, obesity and chronic disease in rural Australia. This requires the consistent use of a conceptual model, such as that proposed by Glanz et al. [17] with its subsequent refinements to encourage researchers to build on previous findings and further our understanding of which constructs are important to interrogate to inform community and policy level change. 

Glanz et al. [17] argue that the community and consumer food environments have a broad impact at a population level and are therefore both of high research priority. Determining the most appropriate methodology to measure these two food environments is vital to inform appropriate interventions to affect change. While there are over 500 known measures to assess food environments there is a lack of reliable, valid and commonly agreed upon methods and little consistency in their application, especially in relation to intervention studies, and particularly for the rural context. The consistent collection of data in a standardised and coordinated manner provides distinct advantages, especially for rural areas, such as the pooling of data where the total number of food stores and outlets per town may be small; then used to inform the development of locally relevant and feasible interventions, potentially enabling the transferability of these local interventions across similar communities; and supporting higher-level advocacy efforts such as food pricing. The INFORMAS network may be a helpful platform in this regard.

The findings of this review do not offer a suite of ‘gold standard’ measurement tools known to be reliable, valid and sensitive to change to assess the community or consumer food environments in rural Australian towns. However, recommendations are proposed to progress this important area of research for the rural context in Australia and other countries. 

## Figures and Tables

**Figure 1 ijerph-16-02416-f001:**
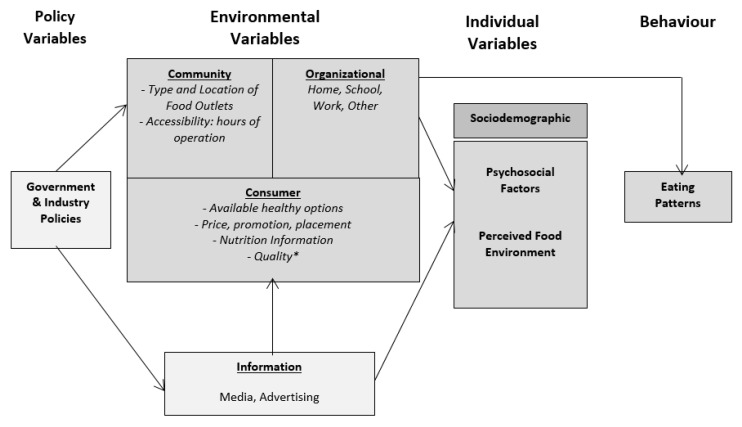
Food Environment Conceptual Model (adapted) [17]. * Quality has also been included as an aspect of the consumer food environment in this review.

**Figure 2 ijerph-16-02416-f002:**
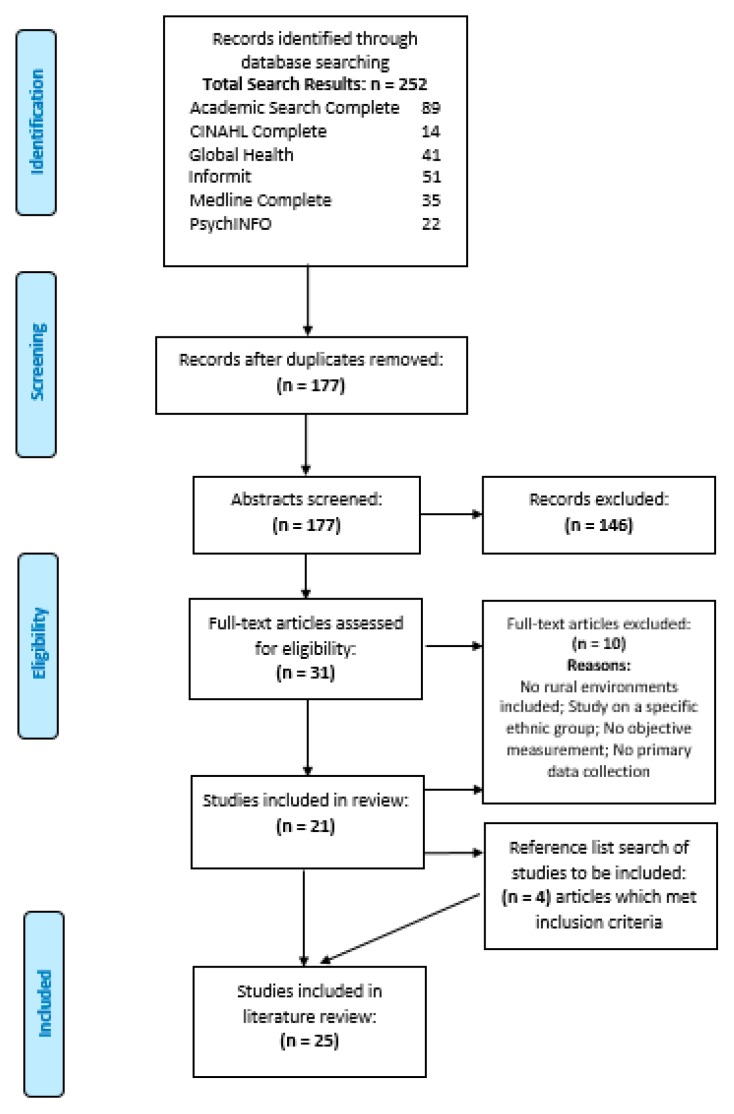
PRISMA diagram of literature search.

**Table 1 ijerph-16-02416-t001:** Overview of studies (n = 25).

Characteristic	n (%)
***Country***	
Australia [22,34,35,36,37,38,39,40,41,42,43]	11 (44)
USA [31,32,33,44,45,46,47,48,49]	9 (36)
Scotland [50,51]	2 (8)
Canada [52,53]	2 (8)
New Zealand [54]	1 (4)
***Context***	
Rural [31,32,34,35,36,40,42,44,45,47,49,52,53]	13 (52)
Mixed Urban/Rural [22,33,37,38,39,41,43,46,48,50,51,54]	12 (48)
***Defined Rurality***	
Yes [22,35,36,38,39,41,42,43,45,46,48,50,51,52,53]	15 (60)
No [31,32,33,34,37,40,44,47,49,54]	10 (40)
***Utilised Conceptual Model***	
Yes [31,32,33,35,36,40,45,46,52]	9 (36)
No [22,34,37,38,39,41,42,43,44,47,48,49,50,51,53,54]	16 (64)
***Study Type***	
Observational [22,32,33,34,35,36,37,38,39,40,41,42,43,44,45,46,47,48,49,50,51,52,53,54]	24 (96)
Intervention [31]	1 (4)
***Study Design***	
Cross-sectional [22,32,33,35,36,37,40,41,42,43,44,45,46,47,48,49,50,51,52,53]	20 (80)
Longitudinal Cohort [34,37,38,39]	4 (16)
Quasi-Experimental [31]	1 (4)
***Psychometric Testing***	
Yes [31,32,33,35,36,38,40,43,46,48,52]	11 (44)
No [22,34,37,39,41,42,44,45,47,49,50,51,53,54]	14 (56)

**Table 2 ijerph-16-02416-t002:** Aspects of the Food Environment Investigated.

Author/Year	Community Food Environment		Consumer Food Environment						Total Food Environment Attributes Researched per Study
	Type & Location of Food Outlet	Accessibility	Available Healthy Options	Price	Promotion	Placement	Nutrition Information	Quality	
^Cuttler et al. (2018) [34]				√					1
^Love et al. (2018) [35]				√					1
^Whelan et al. (2018) [36]			√	√				√	3
DuBreck et al. (2018) [52]	√	√	√	√	√	√	√		7
Larson et al. (2017) [44]			√	√				√	3
^Palermo et al. (2016) [37]				√					1
Byker Shanks et al. (2015) [45]		√	√					√	3
Byker Shanks, Jilcott Pitts & Gustafson (2015) [46]			√	√				√	3
Martinez-Donate et al. (2015) [31]			√	√	√	√	√	√	6
^Chapman et al. (2014) [38]				√					1
Pereira et al. (2014) [32]			√	√	√	√	√	√	6
^Pollard et al. (2014) [22]				√				√	2
^Tseng et al. (2014) [39]	√								1
Pitts et al. (2013) [33]			√	√				√	3
^Innes-Hughes et al. (2012) [40]	√		√						2
^Ward et al. (2012) [41]				√					1
Sadler et al. (2011) [53]	√								1
Sharkey et al. (2011) [47]	√		√						2
Smith et al. (2010) [50]	√		√						2
Wang et al. (2010) [54]			√	√					2
Cummins et al. (2009) [51]								√	1
^Palermo et al. (2008) [42]				√					1
Hosler et al. (2008) [48]	√	√	√						3
Creel et al. (2008) [49]	√	√	√						3
^Herzfeld & McManus (2007) [43]			√	√				√	3
**Total studies researching each attribute**	8	4	15	16	3	3	3	10	

Notes: ^denotes Australian studies.

**Table 3 ijerph-16-02416-t003:** Tools measuring the consumer food environment: Summary.

Assessment Tool/Methodology [Country of origin]	Number of Reviewed Studies Using Tool (References)	Aspect Measured by Tool	Psychometric Properties of Tool
Nutrition Environment Measures Survey - Stores (NEMS-S) * [including NEMS-S-Rev] [USA]	6 [22,31,32,33,36,40]	- Available healthy options - Price - Quality	High inter-rater and test-re-test reliability Good validity in an American context * Innes-Hughes et al. [40] some reliability testing of modified tool * Pollard et al. [22] based quality tool on NEMS-S
Nutrition Environment Measures Survey – Restaurants (NEMS-R)* [including CMA] [USA]	4 [31,32,36,52]	- Available healthy options - Price - Promotion - Nutrition Information	Valid in the American context Acceptable inter-rater reliability Very good test-retest reliability
Victorian Healthy Food Basket (VHFB) [Australia]	3 [37,41,42]	- Price	Not discussed in literature review articles or in original tool development paper [58]
QLD Healthy Food Access Basket Survey (QLDHFAB) * [Australia]	2 [22,38]	- Price - Available healthy options	Chapman et al. [38] tested inter-rater reliability Psychometric properties of tool not discussed in original paper [59]
Healthy Eating Indicator Shopping Basket (HEISB) [USA]	2 [50,51]	- Available healthy options - Quality	Not discussed in literature review paper or in development of tool [60]
Healthy Diets Australian Standardised Affordability and Price (ASAP) Tool [Australia]	1 [35]	- Price	Validity testing described in protocol paper [61]
Quality tool based on NEMS-S [Australia]	1 [22]	- Quality	Developed own quality tool No validity testing Minimal inter-rater reliability testing No test-re-test reliability
Farmers’ Market Audit Tool (F-MAT) [USA]	1 [46]	- Available healthy options - Quality - Price	Face validity reviewed with content experts Inter-rater reliability high Discriminant validity good
Healthy Food Basket – Tasmania [Australia]	1 [43]	- Available healthy options - Quality - Price	Poor inter-rater reliability Test-re-test reliability not tested Validity not tested
Healthy Food Basket [New Zealand]	1 [54]	- Available healthy options - Price	Not discussed

* Includes studies which have adapted the tool.

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
