# Peer review of "Measuring Rural Food Environments for Local Action in Australia: A Systematic Critical Synthesis Review"

_ijerph, 2019, doi:10.3390/ijerph16132416_

Round 1

Reviewer 1 Report

 Overall comments:

The manuscript seeks to investigate food environment measurement tools employed in research in rural Australian towns. While the research is perhaps somewhat marginal (what is the population of regional/remote Australia?), it is an understudied research area, and this manuscript will make a contribution to the field.  

It may be useful to refine/revise the research question: Are the measurement tools employed in rural food environment research appropriate to inform local interventions in rural (outer regional and remote) Australian towns?

The measurement tools studied in the review predominantly assess the consumer food environment, however the review also examines spatial measures (rather than tools) of the community food environment in a rural context. So, it would appear the research question is not just about measurement tools but is also about spatial measures.

The research question also suggests further analysis that will “inform local interventions”. It would be useful to make clear how the authors’ review of consumer food environment measurement tools and community food environments spatial measures will inform local interventions.

While the research question focuses on Australia, it is not clear why the review considered studies from other high-income countries other than Australia.

Abstract

Suggest revising so that the abstract identifies the research gap that this manuscript addresses. Also, a lot of words are on the results, however it is less clear what the conclusion is – i.e. were the measurement tools appropriate?

Introduction

Line 31 -32: reference? Is this the same as the following sentence? The meaning of the sentence – “In low and middle-income countries…” -  is not clear and could be rephrased

Line 44: reference no. 3 is in superscript

Line 51: “relatively new field” compared to what?

Line 54: “Over 500 measures of the food environment exist”. This does not make sense and is somewhat meaningless. Theoretically, an infinite number could exist. Suggest rephrasing.

These references may be useful for the intro:

Wilkins, E. L., M. A. Morris, D. Radley and C. Griffiths (2017). "Using Geographic Information Systems to measure retail food environments: Discussion of methodological considerations and a proposed reporting checklist (Geo-FERN)." Health & Place 44: 110-117.

Lebel, A., D. Noreau, L. Tremblay, C. Oberle, M. Girard-Gadreau, M. Duguay and J. P. Block (2016). "Identifying rural food deserts: Methodological considerations for food environment interventions." Can J Public Health 107(Suppl 1): 5353.

Thornton, L. E., D. A. Crawford, V. J. Cleland, A. F. Timperio, G. Abbott and K. Ball (2012). "Do food and physical activity environments vary between disadvantaged urban and rural areas? Findings from the READI Study." Health Promot J Austr 23(2): 153-156.

Lines 96-97 – Is this suggesting there are no other reviews on measurement tools to assess rural community and consumer food environments worldwide? Or just in Australia?

Methods

Line 120 “developed country”. Previously you referred to high-income country.

Comment on whether participants in the community food environment studies included were adults or children. This will have implication on the food environment measures that are identified.

Results

Line 211: “There were no intervention studies.” Meaning?

Line 225: Define the CBG proximity measure – is this to the centroid? PWC?

Define traditional fast food outlets.

Lines 237 – 240: Your explanation appears to be too definitive. Tseng et al identified a number of limitations to their study and drew several conclusions. It would seem reductionist to draw upon one element, the type of fast food outlet considered, to explain the results. For example, they discussed buffer size as a factor, among others.

Line251-254- ….“two important differences…...” Is this correct? For example, the Tseng study used individual level household addresses for the creation of measures. Therefore, not different to the Sadler study.

Line 254-256: You appear to be confusing two concepts here: i) measures based on road network vs Euclidian distance; and ii) geographic scale, measures that reflect a walkable distance to purchase food in urban areas vs measures that reflect a driving distance to purchase food in rural areas.

Line 266-267: “A weakness of this methodology is….” Not sure what this statement means.

Discussion

Lines 496-500:  â€śâ€¦distinct lack of research…” Unclear why you have drawn this conclusion. Your own review identified 25 studies.  â€śOf the measurement tools assesses in this review, none were appropriate in their current form…” Again, unclear meaning. Is this referring to the consumer food environment tools. Later in the discussion you differentiate between your findings for the community and consumer food environments. The opening paragraph should also reflect this nuance.

Lines 608-611: Unclear what this means.

Conclusion

The authors could comment on the transferability or implications of findings for rural/regional contexts in other countries.

Author Response

Thank you for your review. Attached please find responses to your comments.

Reviewer 2 Report

This is a relevant topic.However, given the overall objective of the paper, in the abstract and the body of the paper, the focus on tools for measuring rural food environment gets lost in the summarization of each study included. The information in the literature needs to be synthesized to address the aims of the paper.  In discussion on each study the authors needs to keep the main objective of the paper in focus.  

Author Response

This is a relevant topic. However, given the overall objective of the paper, in the abstract and the body of the paper, the focus on tools for measuring rural food environment gets lost in the summarization of each study included. The information in the literature needs to be synthesized to address the aims of the paper. 

response:

Thank you for reviewing our manuscript. This review adopted a critical synthesis methodology, therefore it was important that each study included in the review be critiqued in relation to the study aim, namely, their appropriateness for the rural context.  The synthesis of this critique is presented in the results section under each of the relevant community and consumer food environment components.

 In discussion on each study the authors needs to keep the main objective of the paper in focus.  

response: Line 495-497 – This sentence has been altered to strengthen the discussion in relation to the study aim “Of the measurement tools assessed in this review, none were appropriate in their current form to describe or inform local interventions to improve the community or consumer food environments in rural (outer regional and remote) Australian towns.”

Reviewer 3 Report

Methodologically sound. Significant studies have been included in the review. Recommendations are appropriate and relevant for consideration. The focus on rural is needed and the article will support other researchers who focus on rural communities.

Author Response

Methodologically sound. Significant studies have been included in the review. Recommendations are appropriate and relevant for consideration. The focus on rural is needed and the article will support other researchers who focus on rural communities. response: Many thanks for your positive comments on our manuscript.